

# Integrated metabolomic and transcriptomic analysis reveal the effect of mechanical stress on sugar metabolism in tea leaves (*Camellia sinensis*) post-harvest

Zhilong Hao*, Yanping Tan*, Jiao Feng, Hongzheng Lin, Zhilin Sun, Jia Yun Zhuang, Qianlian Chen, Xinyi Jin and Yun Sun

College of Horticulture/Key Laboratory of Tea Science in Fujian Province, Fujian Agriculture and Forestry University, Fujian, Fuzhou, China
* These authors contributed equally to this work.

Corresponding authors
Xinyi Jin, 153340246@qq.com
Yun Sun, sunyun1125@126.com

## ABSTRACT

Sugar metabolites not only act as the key compounds in tea plant response to stress but are also critical for tea quality formation during the post-harvest processing of tea leaves. However, the mechanisms by which sugar metabolites in post-harvest tea leaves respond to mechanical stress are unclear. In this study, we aimed to investigate the effects of mechanical stress on saccharide metabolites and related post-harvest tea genes. Withered (C15) and mechanically-stressed (V15) for 15 min Oolong tea leaves were used for metabolome and transcriptome sequencing analyses. We identified a total of 19 sugar metabolites, most of which increased in C15 and V15. A total of 69 genes related to sugar metabolism were identified using transcriptome analysis, most of which were down-regulated in C15 and V15. To further understand the relationship between the down-regulated genes and sugar metabolites, we analyzed the sucrose and starch, galactose, and glycolysis metabolic pathways, and found that several key genes of invertase (*INV*), α-amylase (*AMY*), β-amylase (*BMY*), aldose 1-epimerase (*AEP*), and α-galactosidase (*AGAL*) were down-regulated. This inhibited the hydrolysis of sugars and might have contributed to the enrichment of galactose and D-mannose in V15. Additionally, galactinol synthase (*Gols*), raffinose synthase (*RS*), hexokinase (*HXK*), 6-phosphofructokinase 1 (*PFK-1*), and pyruvate kinase (*PK*) genes were significantly upregulated in V15, promoting the accumulation of D-fructose-6-phosphate (D-Fru-6P), D-glucose-6-phosphate (D-glu-6P), and D-glucose. Transcriptome and metabolome association analysis showed that the glycolysis pathway was enhanced and the hydrolysis rate of sugars related to hemicellulose synthesis slowed in response to mechanical stress. In this study, we explored the role of sugar in the response of post-harvest tea leaves to mechanical stress by analyzing differences in the expression of sugar metabolites and related genes. Our results improve the understanding of post-harvest tea's resistance to mechanical stress and the associated mechanism of sugar metabolism. The resulting treatment may be used to control the quality of Oolong tea.

## INTRODUCTION

Sugars are the main storage source of carbon and energy in plants. They also act as the key signal molecules to regulate the development and growth of plants by modulating gene expression and enzyme activity (*Kaur et al., 2021*). Specifically, sugars play critical roles in a plant's response to stress, including mechanical stimuli, cold temperatures, osmotic stress, and drought conditions (*Krasavina, Burmistrova & Raldugina, 2014*; *Yue et al., 2015*; *Ma, Dias & Freitas, 2020*). Therefore, changes in sugar content are regarded as the key index to reflect the stress caused by adverse conditions. Under mechanical stress, the accumulation of sugars, including sucrose, glucose, fructose, and galactose, was significant. However, its mechanism remains unknown.

In tea plants, sugars are closely related to the quality of fresh tea leaves and are important for the development of aroma and taste quality (*Cui et al., 2019*). The sugar content has been shown to change during various methods of post-harvest tea processing. For instance, during green tea processing, the sugar content decreased after fixation and drying which facilitates the Murad reaction under high temperature conditions (*Wang et al., 2021*). The withering treatment triggers the increase of $\beta$-amylase (BAM) and $\alpha$-amylase (AMY) activity to promote starch degradation. This results in an increased soluble sugar content and an enhanced development of the sweet taste during the post-harvest processing of white tea (*Zhou et al., 2022*). In Oolong tea processing, Yaoqing (turnover) is a key and unique post-harvest processing treatment involving continuous mechanical stress applied to the leaves to induce the metabolite transformation. This process contributes to the development of the tea's unique aroma and taste quality (*Hu et al., 2018*). Recently, numerous studies using the omics technique have indicated that mechanical stress modulates starch degradation and soluble sugar accumulation during Oolong tea processing (*Huang et al., 2020b*; *Wu et al., 2020*, *2022*). Among these, widely targeted metabolomics have been broadly used to research the post-harvest processing of tea leaves (*Wang et al., 2021*; *Zheng et al., 2021*, *2022*). This process integrates the advantages of nontarget and targeted metabolite detection technologies to achieve high throughput; it also has a high sensitivity and broad coverage (*Chu et al., 2020*). The integrated metabolomic and transcriptomic approaches have been widely used in the studies on abiotically-stressed plants. However, the mechanism underlying the stress-induced regulaiton of sugars metabolism is still unknown.

In this study, Oolong tea leaves were subjected to withering and mechanical stress during post-harvest manufacturing and were studied using metabolome and transcriptome sequencing analysis. The key genes and metabolites involved in the starch and sucrose, galactose, and glycolysis metabolic pathways were comprehensively identified and analyzed. These results will provide useful insights into the sugar-mediated mechanism of mechanical stress in the post-harvest processing of tea leaves. Our results lay the foundation for applying turnover treatments for the quality control of Oolong tea.

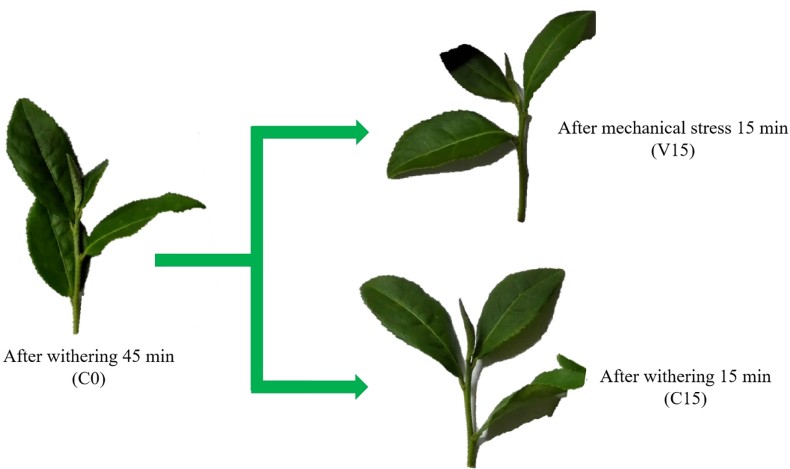

**Figure 1 Changes in the appearance of Tieguanyin tea leaves in V and C groups across the different manufacturing processes.**

# MATERIALS AND METHODS

## Plant materials and mechanical stress treatment

We selected the tea cultivar 'Tieguanyin' (*Camellia sinensis* var. *Tieguanyin*) and harvested it according to the standards of collecting one bud and three leaves. The tea leaves were collected in the garden of Fuqing Nanhushan Tea Co., Ltd. (25°71′N, 119°25′E, Fuzhou, China). The detached tea shoots were divided into two groups. The samples from the first group were treated with continuous turnover for 15 min to simulate mechanical stress treatment (V15); the second group was withered without vibrating stress (temperature $22.0 \pm 0.4$ °C, humidity $83.8 \pm 2$%) for 15 min (C15). The fresh leaves were designed as C0 used to control of C15 and V15 (Fig. 1). Three independent biological replicates were performed for each sampling and the second leaves were immediately collected and frozen in liquid nitrogen and then stored in a −80 °C refrigerator for further analsysis.

The mechanical stress test was performed on a mechanical vibrating Yaoqing bench. The technical parameters of the bench were set as follows: table size $1,100 \times 1,200$ mm, fixed amplitude 25 mm, power 2.2 KW, vibration frequency 330 r/min, maximum load 100 kg, timing control (0–60 min), maximum current 5 A, and voltage 220 V.

## Widely-targeted metabolomics analysis

The metabolite extraction and analysis were conducted by Metware Biotechnology Co., Ltd. (Wuhan, China) (*Chen et al., 2013*). The tea samples were freeze-dried (−50 °C, 10 Pa, 12 h) in a Bisafer-10D vacuum freeze dryer (Saifei Biotechnology Co., Ltd., Nanjing, China) and ground to a powder using a mixing mill (MM 400; Retsch, Haan, Germany) with a zirconia bead for 1.5 min at 30 Hz. A total of 100 mg of powder from each sample was dissolved in 0.6 ml of 70% aqueous methanol solution containing 0.1 mg/l lidocaine as an internal standard and extracted overnight at 4 °C. The mixtures were centrifuged at 10,000 *g* for 10 min to obtain extracts that were absorbed and filtered for UPLC-MS/MS

analysis. The quality control (QC) samples prepared by mixing all samples to check the stability of the instrument.

## UPLC and ESI-triple quadrupole-linear ion trap (Q TRAP)-MS/MS conditions

The aqueous solution was analyzed using an LC-ESI-MS/MS system (HPLC: Shim-pack UFLC Shimadzu CBM30A; MS: Applied Biosystems 4500 Q-TRAP; Thermo Fisher Scientific, Inc., Waltham, MA, USA). The system was equipped with a Waters UPLC HSS T3 C18 column (2.1 × 100 mm, 1.8 μm; Waters Company, Milford, MA, USA) with 4 μL at 40 °C. Acetonitrile (0.04% acetic acid) was used as mobile phase B and acidified water (0.04% acetic acid) was used as mobile phase A. The following linear gradient were used for the eluent B: 5% (0 min), 0–95% (0–11 min), 95% (11–12 min), 95–5% (12–12.1 min), and 5% (12.1–15 min). Further, an ESI-triple quadrupole-linear ion trap (QTRAP)-MS detector (Thermo Fisher Scientific, Inc., Waltham, MA, USA) was used to connect the effluent alternatively. Analyst 1.6.1 software (AB SCIEX Pte. Ltd, Framingham, MA, USA) was used for data acquisition, system, and processing. The ESI operating parameters were: turbo spray; ion source; source temperature 550 °C; ion spray voltage (IS) 5,500 V; ion source gas I, 50 psi; ion source gas II, 60 psi; curtain gas, 25 psi; and the collision activated dissociation (CAD) was set at "high". Quality calibration and instrument tuning were carried out using 10 and 100 μmol/L polypropylene glycol solutions in QQQ and LIT modes, respectively. Decluster potential (DP) and collision energy (CE) were optimized for individual MRM jumps. Monitoring of specific MRM transition sets by metabolites was eluted at each time period (*Chen et al., 2013*).

The metabolites were identified by comparing their retention times, the mass-to-charge (m/z) values, and the fragmentation patterns with those of the authentic standards or by searching against the public databases (MassBank, KNApSAcK, HMDB, MoTo DB, and METLIN). The characteristic ions of each substance were screened by the triple quadrupole, and the intensity of each characteristic ion was detected and collected. Then the chromatographic peak integration and calibration were calculated using MultiQuant software to obtain the relative contents of each substance in the samples (*Fraga et al., 2010*).

## RNA isolation, cDNA library preparation, and RNA-seq

Ten leaves from each sample were randomly selected for total RNA extraction. Total RNA was extracted using the Trizol kit (TaKaRa, Dalian, China) according to the manufacturer's protocol. The sequencing library was prepared using the method of *Huang et al. (2020a)*. After a quality check, the construction of cDNA libraries was carried out according to the protocol for the NEBNext® UltraTM RNA Library Prep Kit for Illumina® (NEB, Ipswich, MA, USA). The sequencing library preparations were created on the Illumina HiSeq 2000 platform (Illumina, San Diego, CA, US) and paired-end reads were determined (2 × 150 bp). The poly-N, adapter and low-quality reads (Qphred ≤ 20) were removed to obtain the clean data and calculate the Q20, Q30, and GC content of the clean reads (Data S1) (*Zheng et al., 2022*).

## Gene annotation and differential expression

The filtered sequences were subjected to the tea plant (*Camellia sinensis* cv. Suchazao) reference genome (*Wei et al., 2018*) localization analysis using Hisat2 v2.0.4 software (*Anders, Pyl & Huber, 2015*). The FPKM cacluation for each gene used HTSeq v0.9.1 as a basis for gene expression identification. Transcriptome sequence data are deposited in the NGDC database (GSA accession: CRA007810 and PRJCA011050).

Differential expression analysis was performed on biological replicate samples using the DESeq2 package (*Love, Huber & Anders, 2014*). Readcount was normalized using DESeq and adjusted for *p*-values using the Benjamini and Hochberg method to control for false discovery rates. All the genes with an adjusted $P \leq 0.05$ as determined by DESeq2 were considered to be differentially expressed. Finally, we predicted the pathways and functions of the differentially expressed genes and made heatmaps using TBtools (*Chen et al., 2020*).

## Quantitative RT-PCR analysis

In order to validate the transcriptome data, 13 DEGs were randomly selected and their expression levels were detected using quantitative RT-PCR (qRT-PCR). The primers were designed using NCBI Primer Blast and listed in Table S1. GAPDH (glyceraldehyde-3-phosphate dehydrogenase) was used as the housekeeping gene (*Zhou et al., 2019*). The total RNA extraction, cDNA synthesis, and quantitative RT-PCR analysis were performed according to our previous study (*Yue et al., 2019*). The qRT-PCR was performed using SYBR Premix Ex Taq™ II (TaKaRa, Dalian, China) and determined on a CFX96 Touch Real-Time PCR System (Bio-Rad, Hercules, CA, USA). The relative expression of the genes was calculated using the $2^{-\Delta\Delta Ct}$ method (*Cao et al., 2020*). Detection was performed in triplicated for each DEG.

## RESULTS

### Variations of sugar metabolites under mechanical stress

In this study, widely targeted metabolomics analysis was performed on C0, C15, and V15. As a result, a total of 19 sugar compounds were identified (Fig. 2 and Table S2). The majority of the sugar metabolites accumulated in C15 and V15.

Compared with C0, sugars including N-Acetyl-D-glucosamine (GlcNAc), glucose-1-phosphate (Glu-1P), D-fructose-6-phosphate (D-Fru-6P), D-galactose, D-glucose-6-phosphate (D-Glu-6P), panose, D-sedoheptulose-7-phosphate, D-sucrose, galactinol, D-melezitose O-rhamnoside, D-melezitose, maltotetraose, and melibiose were significantly increased in C15 ($P < 0.05$). The sugars of GlcNAc, Glu-1P, fructose 6-phosphate-disodium salt, D-Fru-6P, D-galactose, D-Glu-6P, panose, D-sedoheptuiose 7-phosphate, DL-arabinose, and D-mannose were significantly increased in the V15 treatment. Additionally, compared with C15, D-sucrose, galactinol, D-melezitose-O-rhamnoside, D-melezitose, maltotetraose, melibiose, and sucralose were decreased in V15. However, GlcNAc, Glu-1P, fructose 6-phosphate-disodium salt, D-Fru-6P, D-galactose, D-Glu-6P, panose, D-sedoheptuiose 7-phosphate, glucosamine, D-glucose, DL-arabinose, and D-mannose were increased, indicating that the sugar metabolites were regulated by mechanical stress treatment during post-harvest processing.

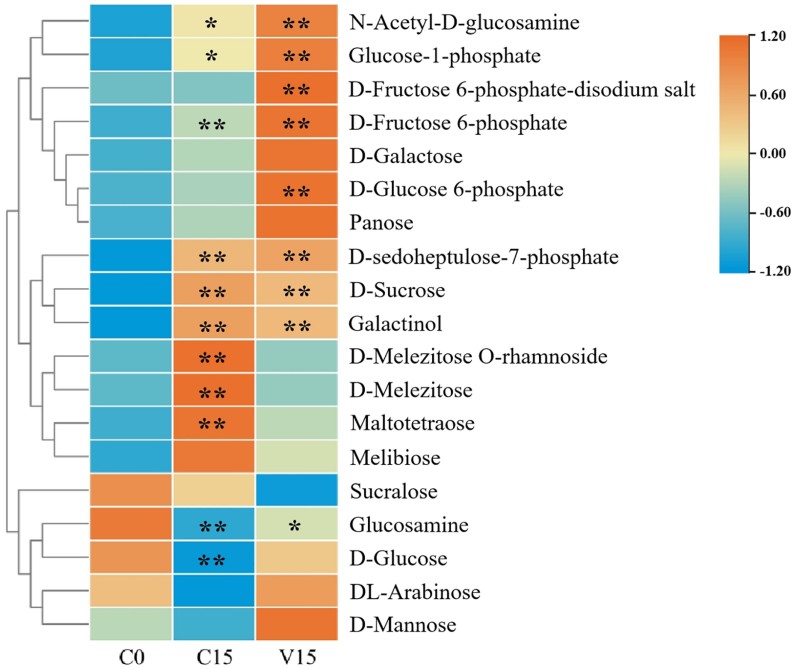

**Figure 2 Heat map of sugar metabolites under different treatments.** Each group was compared to C0. One asterisk (*) and two asterisks (**) represent significant differences at $P < 0.05$ and $P < 0.01$, respectively.

## Differential expression analysis of genes related to the sugar metabolism pathway under mechanical stress

To further explore the transformation mechanism of these sugar metabolites, we performed transcriptome sequencing of the samples. There was a high correlation between the gene expression levels of each sample ($R^2 > 0.95$) (Fig. 3). A total of 13 DEGs were randomly selected for qRT-PCR detection showing that their exression patterns were similar to the RNA-seq results (Fig. S1), indicating good reproducibility and reliability of the RNA-seq data of the tested samples. As showed in Fig. 4, a total of 69 differentially expressed genes related to sugar metabolism pathways were obtained from transcriptome sequencing (FC $\geqq$ 1.2). These genes are associated with the starch and sucrose, galactose, and glycolysis pathways of the sugar metabolism pathway.

Figure 5 shows that the expression of genes related to the starch and sucrose metabolism pathways in tea leaves were down-regulated during post-harvest treatments, particularly under mechanical stress. The down-regulated genes in V15 included invertase (*INV*), trehalose 6-phosphate synthase (*TPS*), trehalose-6-phosphate phosphatase (*TPP*), α-trehalase (*TRE*), phosphoglucomutase (*PGM*), glucose-6-phosphate isomerase (*GPI*), starch synthase (*SS*), 1, 4-α-glucan branching enzyme (*GBE*), *BAM*, and *AMY* when compared to C15. Correspondingly, the sugar metabolites of UDP-glucose, Glu-1P, and D-Glu-6P were increased in tea leaves in response to mechanical stress. In contrast, the up-regulation profiles of hexokinase (*HXK*), UDPglucose 6-dehydrogenase (*UGPase*), and 4-α-glucanotransferase (*DPE*) were associated with the increase of D-Fru-6P, UDP-glucose, and D-glucose in V15.

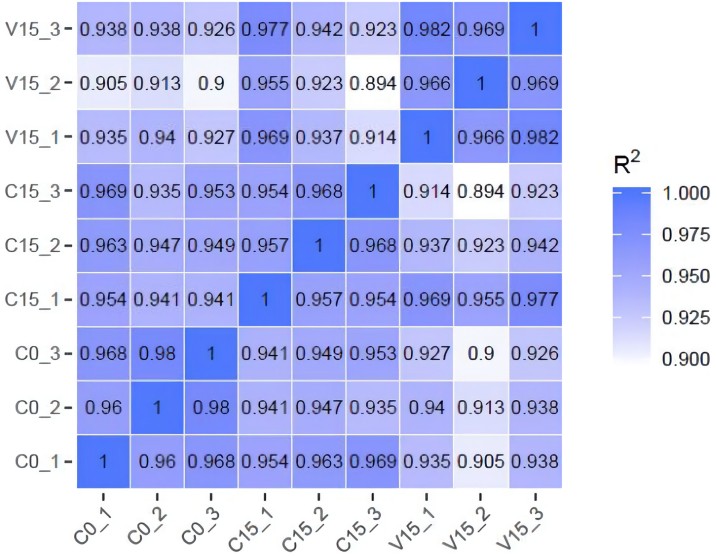

**Figure 3 The heatmap of correlation coefficients between samples.**

In the glycolysis metabolism pathway (Fig. 6), the expression levels of *HXK*, 6-phosphofructokinase 1 (*PFK-1*), phosphoenolpyruvate carboxykinase (*PCK*), and pyruvate kinase (*PK*) genes were up-regulated in V15 compared with C15. The up-regulated expression of these genes reflected the activation of glycolysis in V15. Moreover, mechanical stress decreased phosphoenolpyruvate (PEP) levels in leaves. PEP is an intermediate during glycolysis and its level is inversely proportional to the activity of PFK-1. Additionally, decreasing of PEP could enhance the activity of PFK (*Turner & Plaxton, 2003*).

The key DEGs and metabolites involved in the galacoste metabolism pathway were identified. As showed in Fig. 7, the genes, including aldose 1-epimerase (*AEP*), α-galactosidase (*AGAL*), UDP-glucose 4-epimerase (*GALE*), UDP-glucose-hexose-1-phosphate uridylyltransferase (*GALT*), and *INV* were down-regulated in V15 when compared to C15, and the levels of D-galactose, D-mannose, and D-sorbitol were increased in V15. However, the expression levels of *UGPase*, galactinol synthase (*Gols*), and raffinose synthase (*RS*) were up-regulated in V15, which was related to the increase of D-Glu-6P, UDP-glucose, and D-galactinol levels.

## DISCUSSION

### Adaptive strategies of tea leaf metabolome to mechanical stress

The carbohydrates in plants act as energy and substrate sources for a series of physiological activities. These are mainly derived from the degradation of starch under the catalysis of amylase when facing environmental stressors (*Shen et al., 2015*; *Saddhe, Manuka & Penna, 2020*). The abiotic stress-induced sucrose accumulation in Oolong tea leaves was regarded as a response to adverse environmental conditions. Under these conditions, sucrose is further hydrolyzed into hexose, including glucose and fructose derivatives, *via* the catalysis

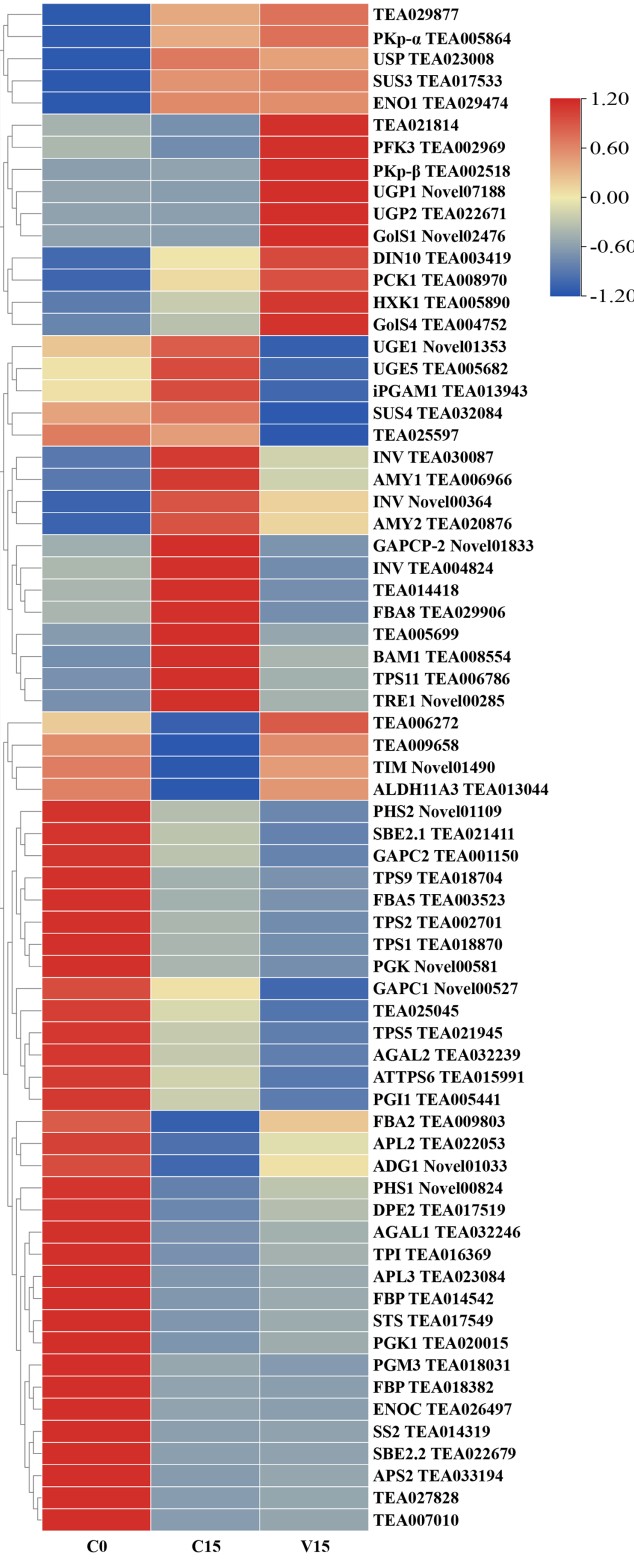

**Figure 4 Heat map of sugar metabolites related DEGs under different treatments.** Based on the hierarchical cluster analysis of fresh (C0), withered (C15), and damaged (V15) leaves, 69 DEGs involved in the carbohydrate metabolism pathway were identified. Red and blue indicate high and low expression on the color bar, respectively.

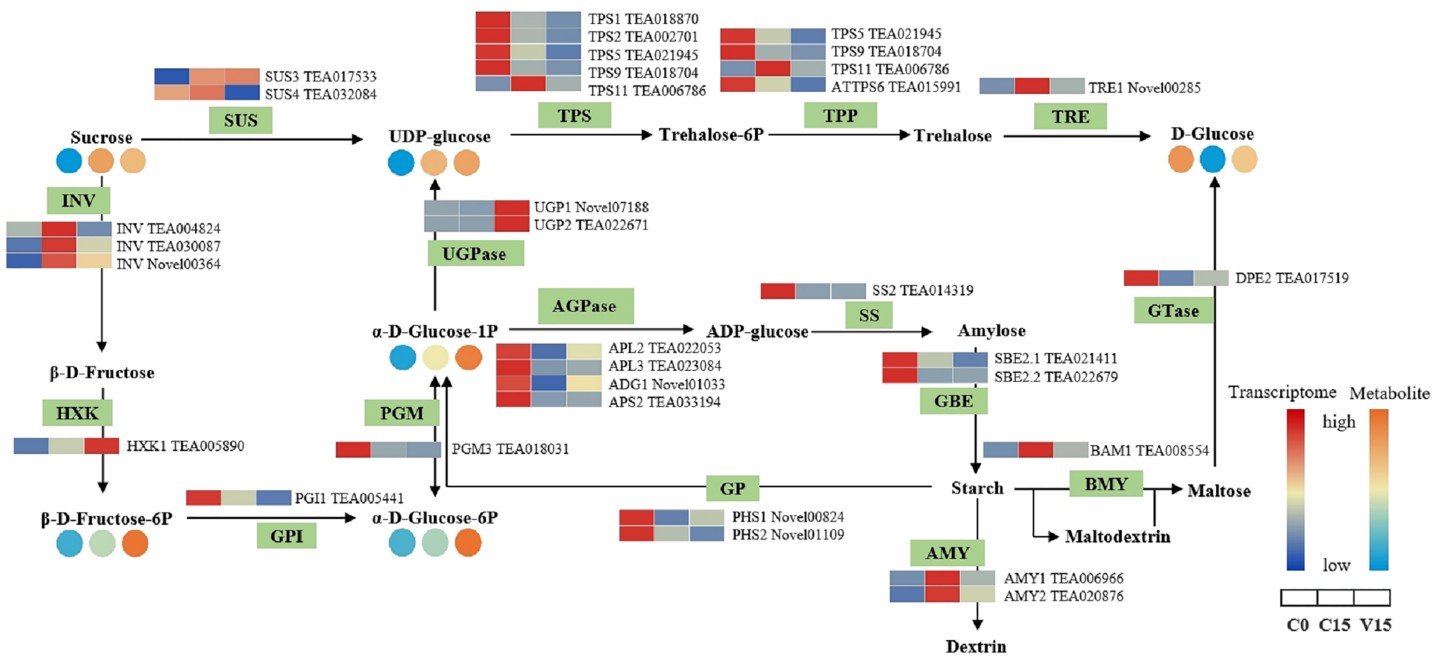

**Figure 5 The metabolites and gene expression levels of sucrose and starch metabolites in leaves under different conditions.** The relative content of each metabolite in leaves is displayed in the form of a heat map from low (blue) to high (orange). Enzymes involved in these pathways were marked in green and the genes encoding the enzymes were put beside them. Similarly, the gene expression levels were denoted in blue (down-regulated) and red (upregulated). Three columns for each metabolite and gene represented groups treated with C0, C15 and V15.

of INV, sucrose synthase (SUS), and HXK, demonstrating a progressive trend in the tea manufacturing process (*Wu et al., 2022*). In this study, the expression of *BAM* genes were dramatically up-regulated after indoor-withering and mechanical stress-enriched glucose levels in V15. Our results are similar to the reports that BAM were involved in the response to abiotic stressors during plant development (*Ma et al., 2022*). The expression of *HXK*, *UGPase*, and *DPE2* genes were higher in V15, and Glu-1P, D-Glu-6P, and D-Fru-6P amount in V15 were more abundant than those of other groups (Fig. 2). This suggests that the monosaccharides accumulated in tea leaves to deal with the stressors, such as dehydration or the mechanical stress from Oolong tea manufacturing. Furthermore, GlcNAc, fructose 6-phosphate-disodium salt, D-galactose, panose, DL-arabinose, and D-mannose were most abundant in V15 because the compounds are elements of the biosynthesis of cellulose and hemicellulose, and are spontaneously involved in the response to mechanical wounding (*Scheller & Ulvskov, 2010*; *Kadokawa et al., 2015*). In addition, our data showed that most of the sugars in Oolong tea increased under mechanical stress, which improved the tea aroma and taste. This is consistent with previous studies that turning-over could improve the aroma and taste of Oolong tea (*Zeng et al., 2020*).
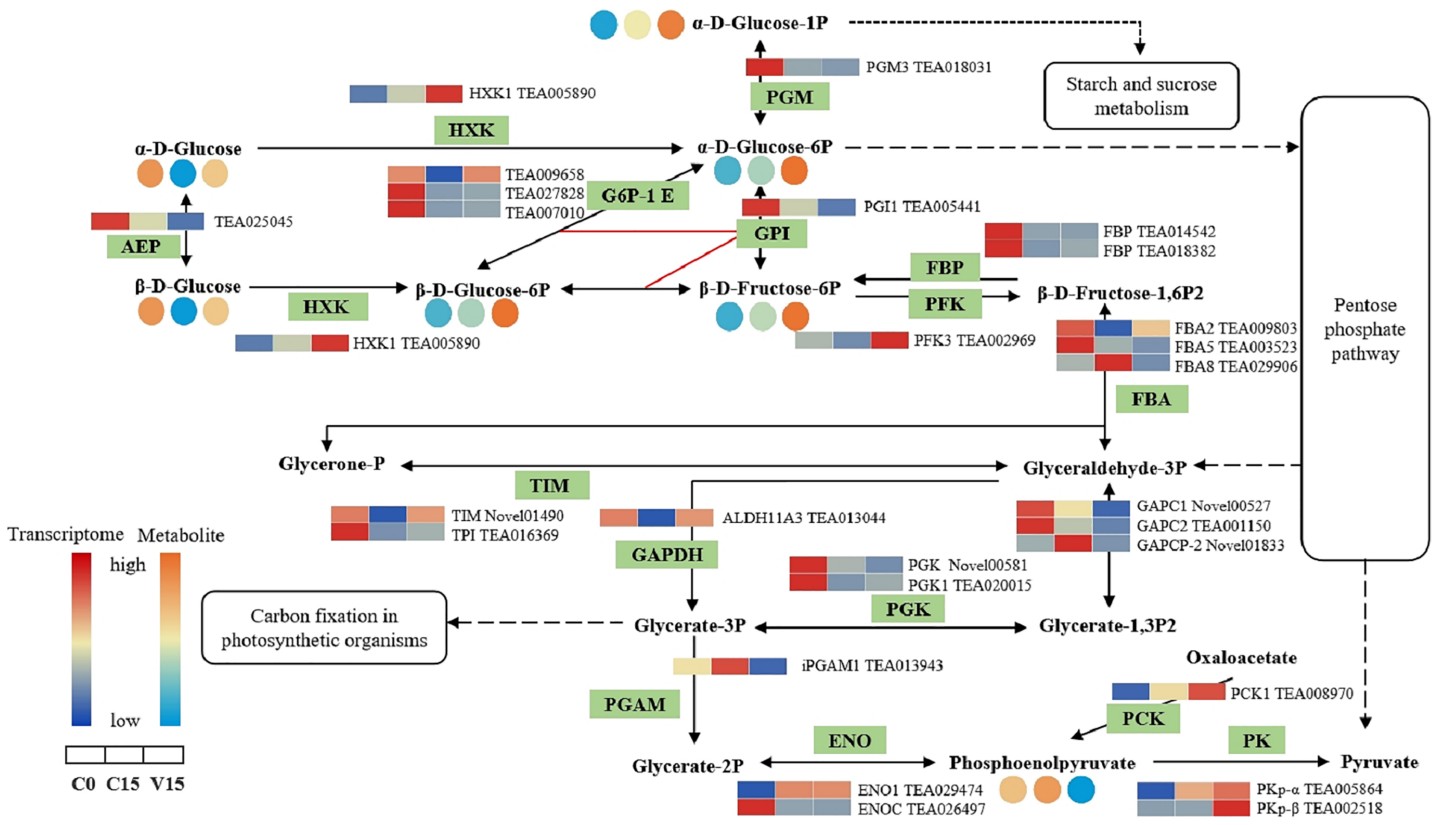

**Figure 6 The metabolites and gene expression levels mapped to glycolysis metabolism in leaves changed under different conditions.** The relative content of each metabolite in leaves was displayed in the form of a heat map from low (blue) to high (orange) as presented in the color scale. Enzymes involved in these pathways were marked in green and the genes encoding the enzymes were put beside them. Similarly, the gene expression levels were denoted in blue (downregulated) and red (upregulated). Three columns for each metabolite and gene represented groups treated with C0, C15 and V15.

## Transcriptome analysis associated with sugar metabolism pathways

### Sucrose and starch metabolic pathway

The INV and SUS have catalyzing roles during the conversion of sucrose into glucose and fructose derivatives. As shown in Fig. 5, the UDP-glucose was significantly enriched in V15, followed by C15 and C0, attributing to the higher expression of *INV* and *SUS* genes in V15 and C15 than in C0. Mechanical stress may give rise to the hydrolysis of sucrose, intermediating into the subsequent synthesis of secondary metabolites for the struggle against adverse environmental conditions (*Ruan, 2012*; *Yang et al., 2018*). Fructose can be further catalyzed into fructose 6-phosphate with the presence of HXK. Notably, the expression of *HXK1* was activated after mechanical stress in this study, contributing to the highest levels of fructose 6-phosphate in V15 than in other groups (Fig. 5). These results are consistent with a previous study that showed that greater HXK activity helps plants improve their ability to cope with different stresses (*Moore, 2003*).

### Glycolysis metabolic pathway

Apart from producing substrate for metabolism, glycolysis also provides energy products including ATP, NADP, and pyruvate *via* carbohydrate oxidation in response to abiotic

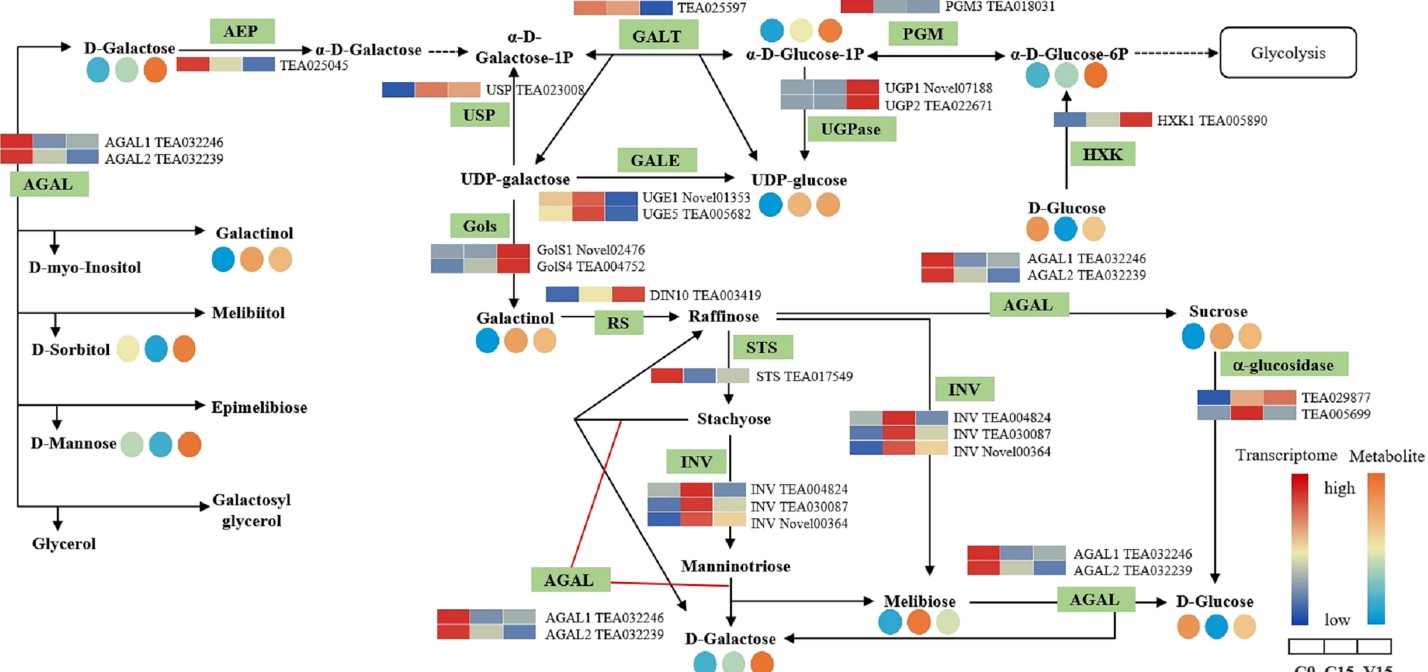

**Figure 7 The metabolites and gene expression levels mapped to galactose metabolism in leaves changed under different conditions.** The relative content of each metabolite is displayed in the form of a heat map from low (blue) to high (orange). Enzymes involved in these pathways were marked in green and the genes encoding the enzymes were put beside them. Similarly, the gene expression levels were denoted in blue (down-regulated) and red (upregulated). Three columns for each metabolite and gene represented groups treated with C0, C15, and V15.

stress (*Gakière et al., 2018*; *Borrego-Benjumea et al., 2020*). As the key enzymes involved in glycolysis process, HXK, PFK, and PK play distinct roles in plant metabolism and development (*Zhang et al., 2011*; *Khanna et al., 2014*; *Peng et al., 2021*). In this study, the expression of *HXK*, *PFK*, and *PK* genes were significantly up-regulated with mechanical stress in V15, leading to the increase of Glu-6P and Fru-6P, and a decrease in PEP (Fig. 6). This observation indicated that glycolysis was enhanced to produce energy in response to environmental stress during the manufacture of Oolong tea. These results support a previous study that showed that glycolysis is activated under stress resulting in increased energy metabolism (*Liu et al., 2019*).

### Galactose metabolic pathway

Galactose, though a non-preferred carbon source, is usually converted to bioavailable molecules including glucose and fructose derivatives (*Kulcsár et al., 2017*). UDP-glucose is catalyzed into UDP-galactose and galactose-1-phosphate in the presence of GALT, then the UDP-galactose would be converted into UDP-glucose under the catalysis of GALE (*Schuler et al., 2018*). In this study, the expression of GALT was decreased in V15, inhibiting the conversion from UDP-glucose to UDP-galactose; while the enhanced expression of UGpase in V15 promoted the accumulation of UDP-glucose (Fig. 7). Moreover, the *Gols* and *RS* genes, associated with the galactinol and raffinose biosynthesis, were up-regulated in V15 (Fig. 7), suggesting that the galactinol and raffinose would be

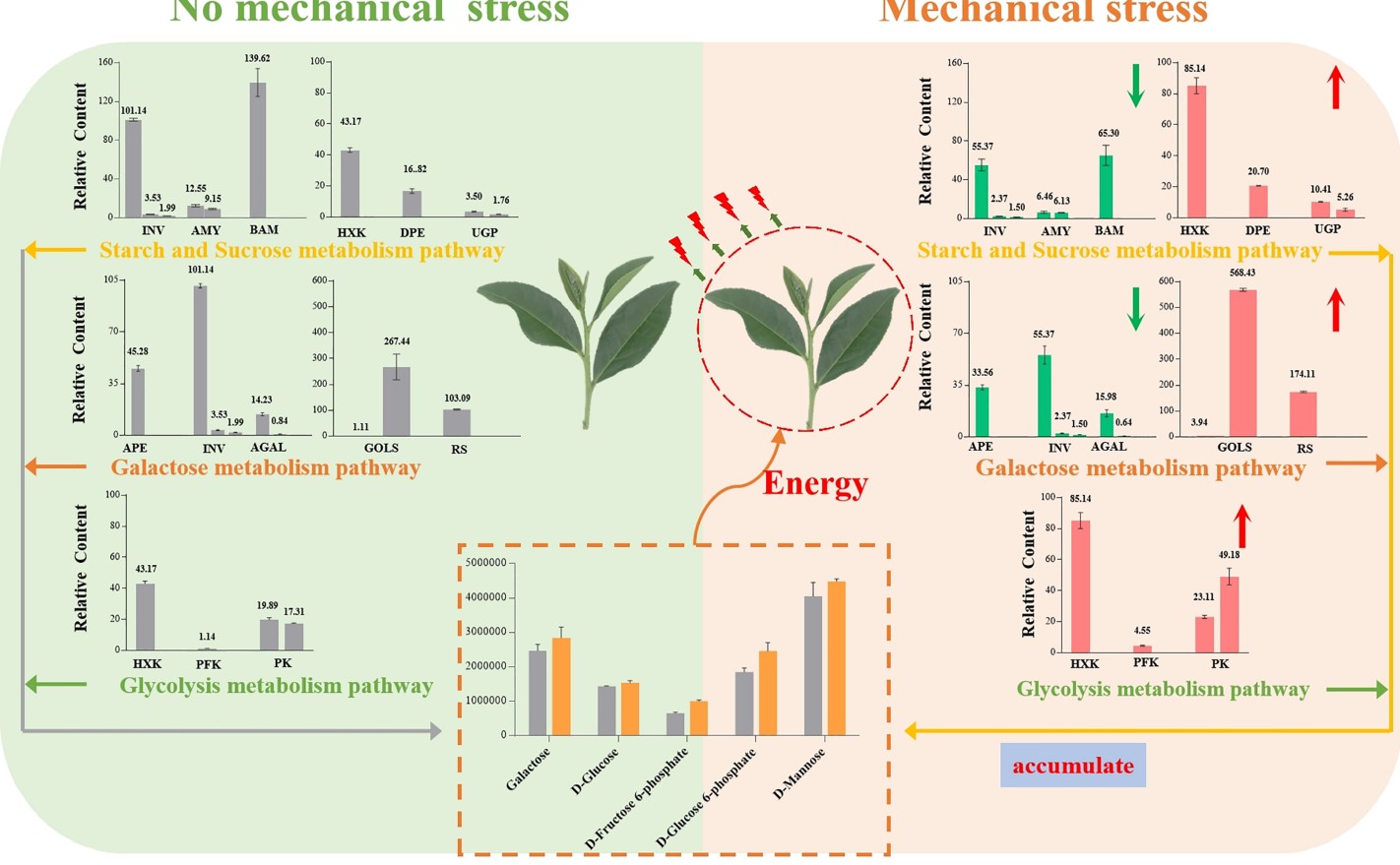

**Figure 8** **Summary of the effects of mechanical stress on the sugar metabolism pathways in Oolong tea (*Camellia sinensis*).** The column diagram represents the expression level of the corresponding gene for each enzyme. Red and green colors indicate up and down expressions, respectively.

induced to accumulate in adverse environments. These results are consistent with previous studies that showed that the accumulation of these two compounds contributed to stress resistance in plants (*Salvi, Kamble & Majee, 2018*; *Li et al., 2020*). As osmotic regulators, galactinol and sorbitol would be hydrolyzed by AGAL (*Chauhan et al., 2012*; *Aulitto et al., 2017*). In this study, the expression of *AGAL1* and *AGAL2* were down-regulated in C15 and V15, retarding the hydrolysis of galactinol and sorbitol (Fig. 7). The spontaneous resistance to environmental stress may be generated in Oolong tea leaves by enriching the osmotic-regulating compounds, which was also observed in previous study (*Zhang, Song & Bartels, 2018*). Therefore, the accumulation of sugars may play an essential role in plant tolerance to abiotic stresses.

## CONCLUSION

Our results showed that Oolong tea leaves enrich sugar metabolites in order to resist mechanical stress (Fig. 8). When Oolong tea leaves were subjected to mechanical stress, the levels of most sugar metabolites increased, and most of the genes related to sugar hydrolysis were down-regulated. Down-regulation of the *INV*, *AMY*, *BMY*, *AEP*, and

*AGAL* genes inhibited the hydrolysis of sugars and was beneficial to the enrichment of sugars in V15, including galactose and mannose related to the stress response. Additionally, a few key genes related to stress resistance were up-regulated in V15, including *Gols*, *RS*, *HXK*, *PFK-1*, and *PK* genes that promoted the production of D-Fru-6P, D-Glu-6P, and D-glucose while enhancing the glycolytic pathway that produces energy. The accumulation of these sugars in tea leaves lays the foundation for the development of tea quality, while providing energy to resist mechanical stress, and helping cells maintain homeostasis. Our results may provide useful insights into the physiological and molecular mechanisms of the response of Oolong tea leaves to mechanical stress and may improve the production process of this tea variety.

### Funding
This work was supported by the Earmarked Fund for China Agriculture Research System (CARS-19), the fujian Province Modern Agricultural (Tea) Industry Technology System Special Project ([2021] No. 637), the fujian Agriculture and Forestry University Tea Industry Chain Science and Technology Innovation and Service System Construction Project (K1520005A06), and the Special Fund for Science and Technology Innovation of Fujian Zhang Tianfu Tea Development Foundation (FJZTF01-FJZTF03). The funders had no role in study design, data collection and analysis, decision to publish, or preparation of the manuscript.

### Grant Disclosures
The following grant information was disclosed by the authors:
China Agriculture Research System (CARS-19).
Fujian Province Modern Agricultural (Tea) Industry Technology System Special Project: [2021] No. 637.
Fujian Agriculture and Forestry University Tea Industry Chain Science and Technology Innovation and Service System Construction Project: K1520005A06.
Science and Technology Innovation of Fujian Zhang Tianfu Tea Development Foundation: FJZTF01-FJZTF03.

### Competing Interests
The authors declare that they have no competing interests.

### Author Contributions
- Zhilong Hao conceived and designed the experiments, performed the experiments, authored or reviewed drafts of the article, and approved the final draft.
- Yanping Tan conceived and designed the experiments, performed the experiments, analyzed the data, prepared figures and/or tables, authored or reviewed drafts of the article, and approved the final draft.
- Jiao Feng performed the experiments, analyzed the data, authored or reviewed drafts of the article, and approved the final draft.

- Hongzheng Lin conceived and designed the experiments, performed the experiments, authored or reviewed drafts of the article, and approved the final draft.
- Zhilin Sun conceived and designed the experiments, performed the experiments, authored or reviewed drafts of the article, and approved the final draft.
- Jia Yun Zhuang conceived and designed the experiments, performed the experiments, authored or reviewed drafts of the article, and approved the final draft.
- Qianlian Chen conceived and designed the experiments, performed the experiments, authored or reviewed drafts of the article, and approved the final draft.
- Xinyi Jin analyzed the data, authored or reviewed drafts of the article, and approved the final draft.
- Yun Sun conceived and designed the experiments, performed the experiments, authored or reviewed drafts of the article, and approved the final draft.

## Data Availability

The data is available at CNBC: CRA007810.

## Supplemental Information

Supplemental information for this article can be found online at http://dx.doi.org/10.7717/peerj.14869#supplemental-information.

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
