# Peer review of "Integrated metabolomic and transcriptomic analysis reveal the effect of mechanical stress on sugar metabolism in tea leaves (Camellia sinensis) post-harvest"

_PeerJ, doi:10.7717/peerj.14869_

## Round 0.1 · original submission · Major Revisions

Kindly address the queries raised by the reviewers.

Reviewer 1 ·

Basic reporting

Professional English is used throughout the manuscript except for some grammatical mistakes such as Pg 4, Line123 - "Remove the poly-N, adaptor..." replace the Remove by "Removed"
Pg 4, Line 132 - "Transcriptome sequences have be deposited"
Please rephrase these sentences.

Experimental design

Authors have used Illumina sequencing data to identify differential gene expression studies. But they have not validated the Illumina data. Authors should validate at least 10 to 15 differentially expressed genes by qPCR analysis or other methods.

Validity of the findings

The authors have deposited the transcriptomic sequences in the NGDC database and provided the GSA accession number but the main concern is that the data is not accessible to viewers.

Annotated reviews are not available for download in order to protect the identity of reviewers who chose to remain anonymous.

Reviewer 2 ·

Basic reporting

See my additional comments

Experimental design

See my additional comments

Validity of the findings

See my additional comments

Additional comments

The authors introduced a healthy background on the role of sugar metabolism in plant response to mechanical damage, focusing on the tolerance and survival perspectives. However, the context of mechanical damage in detached tea leaves is less likely a survival issue rather associated with tea flavor and quality. Unfortunately, the authors largely ignore this important aspect instead they interpreted their results in line with the role of sugar metabolism in repair and stress adaptation mechanisms. In my opinion, lines 44-54 would mislead the readers since these are largely related to general aspects of stress response in intact plants. I would suggest the authors give logical background related to tea processing, such as withering and mechanical damage and associated physiological, metabolic and molecular changes relating to sugar metabolism and tea quality based on the available literature.
Line 19: What do the authors mean by ‘basis and raw materials? please use standard English and technical terms
Line 25: C15 and V15 must be defined in the abstract at the first occurrence
Line 34: what does ‘this’ refer to? Please avoid the use of the confusing pronoun
Line 39: responding? please use standard academic writing
Line 41: What is ‘further the understanding’? Confusing

Line 54: What does tea is a typical plant ..’uses’ mean? The authors should have known the difference between ‘tea’ and ‘tea plant’. Don’t mislead the readers. You should introduce tea plants in a more academic way, such as by mentioning its scientific name. Post-harvest stress does not belong to ‘Tea plant’ rather it is related to ‘tea leaves/shoot’
Line 62: what does ‘this’ refer to? Please avoid the use of the confusing pronoun
Line 66-68: what do you mean by ‘their molecular mechanisms? please use standard academic writing
Line 42: What is ‘turning over’ treatment? How ‘turning over’ treatment could potentially be relevant to sugar metabolism and tea quality should be explained in the introduction and discussion
Signal transduction and stress response are very rapid processes. Harvesting tea i.e. detachment itself is a huge mechanical stress. First, it completely cut off the water supply to the shoot leading to dehydration. Simultaneously, a lot of signaling events rapidly take place in response to the detachment of the shoot. Since mechanical stress was induced 45 min after harvest or detachment of tea shoot, it is the more likely response of acclimated shoot rather than first-hand stress response to mechanical stress. In my opinion, the authors should optimize their presentation.

What are your hypothesis and motivation for this study? What are the specific implications of this knowledge in oolong tea processing?
The authors should also introduce widely targeted metabolomics and its advantages over other methods in the introduction.

Materials and methods
Plant materials and treatment section are missing many important details!
Line 75, 83: How many tea plants/bushes/area (square meters) were used for harvesting? How did you define independent biological replicate?
Line 84: Why did you use the second leaf only for analysis while the tea bud and adjacent first leaf would yield a better quality tea?
Line 84-85, 92: did you freeze dried the tea samples that were previously frozen in liquid nitrogen and stored in below 70C? please explain
Line 90, 98: It seems that Widely-targeted metabolomics and UPLC analysis were not performed in the authors’ own laboratory or institute. If the authors conducted these analyses through outsourcing, they should mention the name of the service provider along with the city and country.
Line 110: Why “Ten leaves from nine samples were randomly selected for independent samples of total RNA”? It is not clear what these nine samples consisted of. Please elaborate and explain logically
It is mandatory to validate RNA seq data and metabolomics data. Please follow standard method for validation.
Fig 1 shows phenotype of tea shoot. Surprisingly, there was no significant visible effect of withering or mechanical stress on tea shoots, even no sign of wilting due to detachment! Please explain.
Line 183: All through the discussion, the authors emphasized plant response to mechanical stress and the role of sugar metabolism. As I mentioned earlier, this is not the relevant context here, the author should focus on tea quality. In line 201, they mentioned an important point that should be expanded in greater detail. How changes in sugar metabolism due to mechanical damage affect tea quality-related metabolites, flavor/aroma and taste are more important than describing how plants respond to stress by altering sugar metabolism. Therefore, I would suggest analyzing important tea quality-related parameters, such as tea polyphenol, amino acids, particularly, theanine, catechins (EGCG), caffeine and so on to logically infer the relation between sugar metabolism and tea quality as influenced by mechanical damage in harvested tea. In addition to biochemical analysis, the authors can also introduce trained tea tasters to score tea quality to better understand the effect of mechanical stress on consumer perception.

Line 311-313: This statement is not clear. Please provide specific details on how this knowledge can be used in tea processing
The manuscript needs professionl proofreading for academic English.

·

Basic reporting

The current manuscript entitled ‘Widely-targeted metabolomic and transcriptomic analyses revealed the effect of mechanical stress on sugar metabolism changes in post-harvest tea (Camellia sinensis)’ explores a very important aspect of carbohydrate metabolism in Oolong tea leaves subjected to mechanical stress. The downregulation of sugar hydrolysing genes under mechanical stress have been proposed as the possible mechanism for accumulation of many saccharides in Oolong tea leading to its characteristic taste. To get the insights in to carbohydrate metabolism, authors have used transcriptomics and metabolomics approaches to establish their findings. Whereas, I find this study interesting but the present form of manuscript suffers many limitations, more specifically the in-depth mining of data has not been done. The “Material and methods’ section, specifically the ‘metabolite extraction’ and instrument parameters used during of LC-MS/MS run are not written properly and need further elaborations.

Line 98-99
It is not clear which mobile phase/solvent system was used during LC-MS/MS run. It is not described whether the isocratic or gradient elution was used. It is also imperative to describe the instrument parameters used by the authors (flow rate, run time, and how the mass spectrum (mass scan mode) was acquired. Merely citing some relevant reference ‘Chen et al. 2013; does not serve the purpose here because Chen et al 2013 described multiple runs on different instruments. Which one the authors have used in this study needs to be clarified. It is also not clear why the solution containing 0.1 mg/l lidocaine was used as the internal standard here. Please explain.
Mention three replicates of each sample in the method section. Were these biological or technical replicates for LC-MS?
As the LC-MS/MS raw data are not shared in any form here, it would be logical to include at least the TIC of the LC-MS run of samples and QC run in the form of supplementary material in this manuscript.

Line 116-128
The important information about the method of library preparation is missing with regard to what was the size selected for library preparation. Other descriptions are also missing like what was the read length of the generated reads? Whether the single-end or paired-end reads were generated. Why the reads with a Phred quality score of (Q<20) were used for downstream analysis, despite the fact that most samples are having about ~90% reads with a Phred quality score of (Q<30).

Line 146-149, The results presented need to be revised. what is meant by the significantly increased? Which statistics are the basis to say this? Presenting this information in the form of a Heatmap obscures many important aspects. As there are only three sample groups to compare, It would be better if this data is presented as a comparison between ‘C0 vs. C15’, and between ‘C0 vs V15’ and also ‘C15 vs V15’ the results are presented in form of volcano plots along with the fold changes (FC value) and p-value o genes and metabolites.
Line 161-163 and Figure 3 How does the high correlation indicates reproducibility and reliability of transcriptome data? The results shown in figure 3 are misquoted here and a wrong interpretation is presented.
Instead of presenting the finding separately in the form of Figures 5, 6, and 7 for starch and sucrose metabolites, galactose metabolism, and glycolysis metabolism, the possibility of presenting it in a single integrated pathway map showing the common substrates and connecting metabolites should be explored to present an integrated view.

There are other syntax errors. A non-exhaustive list of a few is given below: -
Replace ‘oolang tea’ with ‘Oolang tea’ throughout the manuscript.
Expand the abbreviations specifically ‘enzyme names’ on their first use in the text and subsequently abbreviations can be used.
Line 117-118: Rephrase the line to convey correct meaning.
Line 132 Replace ‘have be’ with ‘have been’
Line 217-218:- Rephrase the heading

Experimental design

Experimental design is fine for this study.

Validity of the findings

Its covered in the study. Data presented are validated at two Omic levels.

---

## Round 0.2 · Minor Revisions

Kindly address queries raised by reviewer 1.

Reviewer 1 ·

Basic reporting

No comment

Experimental design

The authors have not mentioned how they validated the transcriptome data in the quantitative RT-PCR analysis section. How they extract the RNA and at which stage they have isolated it for validation. Whether they have been given DNase I treatment or not. Likewise, which methods are used for cDNA synthesis? Authors can use the following articles as a reference to know how to write the quantitative RT-PCR analysis:
1. Kumar, N., Tokas, J., Raghavendra, M. and Singal, H.R., 2021. Impact of exogenous salicylic acid treatment on the cell wall metabolism and ripening process in postharvest tomato fruit stored at ambient temperature. International Journal of Food Science & Technology, 56(6), pp.2961-2972.
2. Mann, A., Kumar, N., Kumar, A., Lata, C., Kumar, A., Meena, B.L., Mishra, D., Grover, M., Gaba, S., Parameswaran, C. and Mantri, N., 2021. de novo transcriptomic profiling of differentially expressed genes in grass halophyte Urochondra setulosa under high salinity. Scientific Reports, 11(1), pp.1-14.

Validity of the findings

No comments

Additional comments

Authors need to format (justify) the introduction part also. Else manuscript is acceptable in its present form

·

Basic reporting

No comment

Experimental design

No comment

Validity of the findings

no comment

Additional comments

The authors have made a good attempt to revise the manuscript. I really appreciate the efforts of the authors toward validating the 13 DEGs using qPCR. My most of the concerns have been addressed in the revised version. However, authors have conveniently skipped a few suggestions, possibly because it may have involved a significant rewrite of the result section. Still, I would suggest seeing the heading "Transcriptome analysis associated with sugar metabolism pathways synthesis.
sucrose and starch metabolic pathways" what is meant by 'sugar metabolism pathways synthesis' please recheck. (Line 339-340 of track change version)
Line 163 (Check Leves should be replaced, its an typo)
I appreciate the efforts of the authors in revising the MS.

---

## Round 0.3 · Minor Revisions

Robert Winkler, the Section Editor, has commented and said:

"Please proofread the manuscript. Already in the first sentence, a verb is missing: " Sugar metabolites not only act as the key compounds in tea plant response to stress but [ARE] also critical for the tea quality formation during post-harvest processing of tea leaves."

Parenthesis missing, unclear number format, e.g., 1,100 mm ◊ 1,200 mm), etc.

Please specify the criteria for identifying metabolites with MS."

---

## Round 0.4 · accepted · Accept

Having gone through the manuscript, the authors have made significant changes in the manuscript. I think this manuscript is technically sound enough for publication.